# Telerehabilitation in the Transitional Care of Patients with Sequelae Associated with COVID-19: Perception of Portuguese Nurses

**DOI:** 10.3390/ijerph192417096

**Published:** 2022-12-19

**Authors:** Neuza Reis, Maria José Costa Dias, Luís Sousa, Inês Agostinho, Miguel Toscano Ricco, Maria Adriana Henriques, Cristina Lavareda Baixinho

**Affiliations:** 1Nursing Research, Innovation and Development Centre of Lisbon (CIDNUR), 1900-160 Lisbon, Portugal; 2Centro Hospitalar Universitário de Lisboa Central (CHULC), 1169-050 Lisboa, Portugal; 3Higher School of Atlantic Health, 2730-036 Barcarena, Portugal; 4Portugal Comprehensive Health Research Centre (CHRC), 7000-811 Evora, Portugal; 5Nursing School of Lisbon, 1900-160 Lisbon, Portugal; 6NOVA Medical School|Faculdade de Ciências Médicas (NMS|FCM), 169-056 Lisboa, Portugal

**Keywords:** COVID-19, pandemic, long COVID, transitional care, eHealth strategies, nursing, rehabilitation

## Abstract

The COVID-19 pandemic brought many changes and challenges to health professionals, due to a lack of knowledge when dealing with the disease, fear of contagion, and the sequelae that characterize long COVID. To deal with this situation, respiratory rehabilitation programs are recommended in face-to-face and/or telerehabilitation modalities. (1) Background: This study had as its primary aim identifying the aspects/components to be considered in the planning and implementation of telerehabilitation interventions that guarantee transitional care for people with long COVID-19 after hospitalization and as a secondary aim identifying the positive aspects of telerehabilitation. (2) Methods: The method used to answer the research question was a focus group, carried out online with eight nurses specialized in rehabilitation nursing. The answers to the semi-structured interview were subjected to content analysis, and qualitative data analysis software (WebQDA^®^) was used to organize and analyze the findings. (3) Results: Four categories emerged from the content analysis: coordination between care levels; transitional care telerehabilitation intervention; advantages of telerehabilitation; and opportunities. (4) Conclusions: These findings make an important contribution to the reorganization of transitional care, allowing the identification of central aspects to be considered in the planning and implementation of telerehabilitation programs for people with long COVID.

## 1. Introduction

The coronavirus disease (COVID-19), caused by the severe acute respiratory syndrome coronavirus-2 (SARS-CoV-2) [1], is transmitted by close contact between people and by the exchange of secretions, such as coughing, sneezing, tears, and saliva [2]. Patients with COVID-19 present as their main symptoms fever, cough, productive cough, dyspnea, odynophagia and nasal congestion, myalgias, chills, headaches, nausea or vomiting, and diarrhea [1].

The prevalence of people who, after the disease, have chronic and persistent symptomatology is high, even after they test negative for COVID-19, with this situation having an impact on daily life activities and, consequently, on the return of these people to their daily lives [3]. According to the World Health Organization (WHO) (2020), when symptoms persist for up to three months after infection, lasting at least two months and without an alternative diagnosis, the patient is diagnosed with “post-COVID-19 condition” or “long COVID”. However, this classification is still not consensual in terms of definition and standardized nomenclature, which makes its diagnosis difficult [4].

The main signs and symptoms of long COVID are dyspnea, fatigue, muscle weakness, lung function abnormalities, decreased quality of life, lung radiological changes, reduced exercise tolerance, restricted lung volumes, impaired resistance, and diffusion, in addition to a decrease in the distance covered in the six-minute walk test [5,6,7].

The authors note that it is difficult to predict the number of people with the disease who will progress to long COVID, as the incidence and mortality rates of this pathology vary by country [7,8]. However, studies have shown incidence rates of long COVID of 76% of people at 6 months [7], 32.6% at 60 days [9], 87% at 60 days [10], and 96% at 90 days [8]. Available data have also shown that the sequelae also affect those who did not have severe cases of the disease [8].

Post-COVID-19 functional impairment affects patients’ ability to perform daily life activities and their functionality, alters their professional performance, and makes social interactions difficult [11]. In addition to the pathology itself, prolonged hospitalization (with or without mechanical ventilation) leads to pulmonary, cardiovascular, muscular, and cognitive changes, as well as anxiety and depression [12]. Many of these patients are able to recover pre-COVID functional levels within the first six weeks, spontaneously or with minimal, preferably remote, support [13].

A recent report presented by WHO in the long COVID context presented data on a workload reduction of 45% in people with long COVID, and 22% did not return to work [14]. Due to the sequelae and their impact on the prognosis, evolution of the respiratory condition, functional capacity, and life quality, rehabilitation after discharge is recommended, with emphasis on respiratory rehabilitation programs and increased tolerance for exertion to be maintained over time.

In view of this scenario, WHO (2021) warned of the need to rethink and reorganize the provision of health care, with a transdisciplinary approach to the assessment, management, and training of people with long COVID, allowing improvement of clinical results, greater patient satisfaction, and ensuring the sustainability of health systems. Considering that COVID-19 presents a high risk of contagion, social distancing is strongly recommended, which makes traditional outpatient rehabilitation difficult [14].

The big issue was to maintain continuity of care from hospital to community that allows an integrated transitional care rehabilitation program. Transitional care includes interventions between the hospital and the community, in three distinct stages: before the person leaves the hospital; at the time of hospital discharge; and, finally, the chronological arc between forty-eight hours and thirty days after discharge [15].

Other studies observe that e-health and telerehabilitation [16,17] are a growing field that can improve access to care and improve health outcomes such as preventive health; management of chronic pain, anxiety, and depression; and rehabilitation-related interventions [17]. The benefits are also salient when discussing transitions of care from inpatient care to community settings, with advantages in reducing length of stays, preventing re-hospitalizations, and promoting better disease management [17].

Faced with the above, telerehabilitation has emerged as a response to resolve this difficulty in the rehabilitation process for post-COVID-19 patients and reduce risks [11]. This makes it possible, through the use of new telecommunications technologies, to provide safe and supervised rehabilitation care in the sick person’s home or anywhere else, in real time or not, conferring advantages similar to those of rehabilitation with face-to-face supervision and minimizing barriers of distance, time, costs, and risks. The programs are essentially based on training for respiratory disease management and physical exercise [13].

In view of the above, this study had as its primary aim identification of the aspects/components to be considered in the planning and implementation of telerehabilitation interventions that guarantee transitional care for people with long COVID-19 after hospitalization and as a secondary aim identification of the positive aspects of telerehabilitation.

## 2. Materials and Methods

The method chosen for this qualitative study was a focus group, as it allows the collection of data from a group with similar experiences, allowing interactions in group discussion based on the researcher’s active role in stimulating the group’s discussion for data collection purposes [18]. Due to its characteristics, this method was suitable to answer the research question “Which telerehabilitation interventions guarantee transitional care for people with long COVID-19 after hospitalization?”

Since the rehabilitation of the sequelae of COVID-19 is a new event, we consider it important to explore the experiences of rehabilitation nursing professionals involved in providing care to these people.

The study protocol complied with the following phases: planning, preparation, moderation, data analysis, and dissemination of results [18,19].

The eligibility criteria were: being nurse specialists in rehabilitation nursing in hospital and community contexts, with clinical experience in care transition, telerehabilitation in the transition process and/or telemonitoring of rehabilitation programs for post-infection by SARS-COV-2. Given that in Portugal there are few nurses involved in digital health projects, we asked colleagues from the Portuguese Association of Rehabilitation Nurses and the specialty college of the Order of Nurses to help identify specialists with the mentioned inclusion criteria. Following the recommendations of Krueger and Casey (2014), the sample was intentional and homogeneous to allow focusing of the discussion on the topic, since the participants had in common a relevant characteristic regarding the topic under discussion—having participated in telerehabilitation programs that guarantee the continuity of care between hospitals and communities [19].

After reviewing the literature on the subject, the interview guide was organized around the following question: In your opinion, how should telerehabilitation interventions for people with post-hospitalization care needs for COVID-19 be organized? From the initial question, it was possible to “taper down” to more specific questions [19]. The central question, as well as the secondary questions, had the purpose of leading the focus group, without conditioning the participants’ sharing. It was foreseen that more specific questions would arise depending on the group’s contributions. This approach of commitment to the group discussion makes it possible to access, on the one hand, the perspectives of the participants themselves in the first part of each discussion and, on the other hand, the answers to the researcher’s specific interests in the second part [18].

The focus group was conducted online, using the Colibri^®^ platform, on 28 April 2022; it lasted 2 h and 25 min, and was recorded and transcribed. The choice of moderator and co-moderator was based on the authors’ recommendation [18,19] and made it possible to increase rigor. The moderator led the discussion, and the co-moderator handled the recording equipment and controlled the logistical conditions [19].

The recorded data were transcribed by one of the researchers in order to visualize what happened in the group [18]. The qualitative analysis of the findings was carried out by two researchers independently, following three steps: (1) data organization; (2) categorization of data with listing of central ideas and regrouping by similarity; and (3) data interpretation.

In defining the categories, representativeness, exhaustiveness, homogeneity, and relevance were ensured [20]. The analysis of the findings was carried out using computer software (WebQDA^®^) (webQDA | Análise Qualitativa Suportada por Software, Ludomédia, Oliveira de Azeméis, Portugal), which enabled the organization and analysis of the findings and increased rigor.

The project was authorized by the Ethics Committee of the Centro Hospitalar Universitário de Lisboa Central (Ruling No. 1209/2022 of 03/18/2022).

All participants freely and clearly agreed to participate in the study by signing the consent form. The data obtained were kept confidential; each participant was assigned a letter—P—and a number (1, 2, 3, …).

The focus group data (recording and transcribed text) were stored securely by the main investigator, not allowing access by others.

## 3. Results

The participants in the focus group were eight nurses, mostly women (n = 5), with an average age of 46.5 years, and had been working, on average, for the last 27 years. They carried out their professional activities in mainland Portugal (n = 4); on the island of Madeira (n = 1); and on the islands of the Azores archipelago (n = 3). Four were working with clinical practice in primary health care, and four were working in a hospital environment.

In the content analysis, the following categories emerged: coordination between levels of care (registration units [UR] = 33); transitional care telerehabilitation intervention (UR = 39); advantages of telerehabilitation (UR = 29); and opportunities (UR = 15) (Figure 1).

### 3.1. Coordination between Levels of Care

Regarding coordination, the participants observed that the emergence of a new pandemic about which there was no knowledge forced learning of care to occur simultaneously with knowledge building about the disease, its evolution, its prognosis, and rehabilitation. The need to provide care, while knowing so little about the disease, was a challenge that generated a lot of insecurity in health professionals about ensuring continuity of care for patients who were hospitalized in intensive care units and COVID services. After a period of hospitalization that was more or less long, some patients were left with sequelae that did not allow or hindered the performance of daily life activities when they returned home.

Post-COVID sequelae and their impact on the functionality of patients, throughout the life cycle, have led to new “ways of doing” to ensure continuity of care, through coordination between hospital nurses and nurses in primary health care. One participant observed that:


*One thing is the environment when we are in the hospital, another thing is home, feeling and being there. (P3)*


The transition from health to illness was experienced with great fear by individuals and families. In addition, difficulties in returning home associated with the worsening of symptoms (dyspnea, cough, myalgias) in the execution of some activities were understood by patients and families as relapses of the disease or worsening sequelae, leading people to stop rehabilitation programs.

As the nurses corroborated in their comments, coordination of care was fundamental and had to be carried out:


*For the coordination of the different services, for the coordination of our part with the families, and with the users. (P1)*



*Our patients who are hospitalized, there is a first contact, a service orientation for these patients, where we often even make a home visit to assess residual architectural barriers and the knowledge of the family, while patients have not yet been discharged. (P3)*



*The person is the same, but why is then seen as a hospital staff or a health center staff. In reality, we are all participants in the sick person’s process, and it is necessary to face exactly that. It is this connection of greater proximity, of non-rupture and separation between the places where we are, and we understand precisely, how we are participating in the life of that patient, in this case it is evolution from a disease process to a health process, which is our goal, and at different times. (P7)*


For one of the participants, what is important is this coordination of proximity with support for the patient and family, and the fact that:


*Establishing this referral circuit for the health care unit in the community (…), with the patient still in hospital, makes sure that with this proximity with the patient is not lost. (P3)*


Communication and pre-discharge coordination is seen as an important contribution to preventing defragmentation of continuity of care and interruption of the rehabilitation program started in the hospital, but was also seen as a major flaw in some contexts:


*Create the referencing and forwarding systems and that’s the big flaw, regardless of whether we now have new technologies. We should use them more and more to share knowledge and transfer this information that is so important to give continuity of care and maintenance of this care, which does not exist. (P6)*


There was a consensus that this coordination should be multidisciplinary, with definition of a transitional care program that guarantees the continuity of interventions initiated in the hospital, with nurse specialists in rehabilitation nursing being the professionals best prepared to manage this transitional care.

Transitional care, in an e-health modality, implies communication with primary care colleagues, who must visit families, even with hospitalized patients, with the dual purpose of assessing family motivation, needs, and abilities to play the role of informal caregiver, and understanding the concerns/fears associated with discharge and the course of the disease, even with regard to unfounded fears of contagion of families.

This identification and characterization of families and community resources aims to minimize disabilities and promote knowledge and acquisition of skills by clients and caregivers to adapt to the new health condition.

### 3.2. Transitional Care Telerehabilitation Interventions

Regarding transitional care telerehabilitation interventions, the experts considered that it is possible to ensure all interventions in an e-health modality, but they suggest that in order to guarantee success in the transition from hospitals to communities, it is necessary to foresee moments of face-to-face assessment of the specific care practice environments that are the patients’ homes.

As these nurses remarked:


*So, for planning it is essential to have direct contact with these people, with these families. To understand what their health problems are, at that moment, what their expectations are, what their adherence to programs is, such as telekinesis, so that we can meet the needs of the patients and especially the families, right?… Evaluate the patients and even establish an agreement with the families, what they want, what they expect from us, what you want, what you want to improve in a certain way, what the goals are, what you want to do again, what I think you can’t do, that is, I think this planning should be done face-to-face. (P1)*



*What I read and realized, but which is also common, precisely, to the projects we had here, is that telerehabilitation must not be separated, it cannot be isolated, there must always be a face-to-face moment, even if it is, precisely, with a home visit (…) I think that this assessment at patients’ homes should include not only the assessment of the patients, but also of the homes, of the people’s social conditions, which includes families and caregivers. (P3)*


The need for face-to-face moments is also important for the assessment of the specific programs for respiratory and motor rehabilitation of people with sequelae:


*There are techniques that are lost, the evaluation is more subjective. It is true that they have the oximeter that allows us to assess oxygen saturation, heart rate, respiratory rate, but there are things that are lost, another type of assessment that we already know. Even auscultation, which is essential for us to make an objective assessment of the patient (P1).*



*We always do an initial assessment. The initial assessment has to and should be face-to-face and, in addition to being face-to-face, it must inspire and include what the colleagues said (relationship of trust and agreement with patients and families regarding the programs). (P6)*


These programs should focus on respiratory rehabilitation, with exercises that increase ventilatory capacity, thoracic expansion, diaphragm performance, control of associated symptoms (cough, dyspnea and expectoration), and exertion tolerance.

Motor rehabilitation should promote gains in terms of muscle strength, flexibility, joint range of motion, and improvement in gait capacity and quality. Changes associated with immobility during the hospitalization period emerged in the professionals’ discourse as sequelae that, in some cases, were not directly related to the infection, but rather were related to the restrictions that contingency plans and the organization of hospital care placed on the mobility of patients in services, and that have an impact on the ability to walk, dyspnea, and quality of life quality after discharge.

The concern with the empowerment of families also stands out, since visits to services were limited to hospital inpatient services and continue to be in services with patients with active infection by SARS-CoV-2. The absence of caregivers is one of the reasons for planning face-to-face moments in telerehabilitation, as mentioned above.

It should also be noted that nurses considered that with a mixed modality of intervention, face-to-face and e-health, the results obtained demonstrate improvement in respiratory, functional, and self-care capacity:


*In terms of results, I can tell you, and compared to the program we have in the gym, the gains are exactly the same, nothing takes away the face-to-face, they are just supplements. (P4)*


### 3.3. Advantages of Telerehabilitation

The aforementioned results reinforce that transitional care guaranteed in an e-health modality is a clear opportunity for nursing care in general, and specifically for rehabilitation nursing.


*The indication was widespread, we are going to reduce the intervention in respiratory rehabilitation, but we did not want our users, our children, to be harmed in any way. And that’s when we started respiratory telerehabilitation (…) for situations of worsening of the respiratory disease, the risk of hospitalization was very high. All the other children who were stable, those who had already mastered the breathing exercises, or whose caregivers were already trained, we started to provide support, then, through video consultations. (P8)*


The emergence of the pandemic crisis, and the need to reorganize care and reallocate resources, took human resources away from communities and from proximity care. The risk of contagion for the most vulnerable populations, the elderly and the chronically ill, and the scarcity of personal protective equipment and human resources, strongly influenced the possibility of carrying out home visits for surveillance, monitoring, treatment, and/or rehabilitation of people with chronic illness, disabilities, and dependence in loco in their life contexts. Although some telerehabilitation solutions already exist in the country, the professionals themselves confess that they looked at them with some suspicion, due to ignorance or prejudice, in relation to their effectiveness, as they do not allow direct contact with people.

This experience changed the attitudes of professionals, and their adherence to digital health solutions:


*There is a reduction in the number of infections here, the situation is improving, at least for now. We considerably increased our face-to-face activity, home visits, but here as a team we decided that respiratory telerehabilitation is to be maintained as a complement to face-to-face intervention. (P8)*



*I just want to add one more thing here. Telekinesiotherapy (…) is a huge challenge. It has allowed us to grow as nurses, to adapt to the new reality in order to be able to give greater support to our users. (P1)*



*The program only appeared a year ago, and it was a complement to the rehabilitation program that we have in an outpatient context… in hospital and the program are exactly the same as that of kinesiotherapy (…) We had already thought about starting at the time and the pandemic came to give that final push. (P4)*



*I believe that telerehabilitation, telemonitoring, telehealth is the future, whether you like it or not, and that is what will be imposed. My reality is that, as I see more and more patients, they are increasingly chronic, with greater demand for care follow-up, at different levels and for longer (…) they had mobility all their lives, and then they come with situations of very difficult mobility, but they always, always come, until the end of their lives. Of course, there were things we could do, this is not all but a part. For certain patients, telerehabilitation, in some family and housing contexts, can be a little difficult to implement, but in other cases it would be a comfort. (P2)*


### 3.4. Opportunities to Telerehabilitation

This opportunity was also seen to create the possibility of improving care for people with prolonged health conditions/illnesses who need healthcare support for long periods of time, reducing trips to emergency services and avoiding unnecessary travel.

It can be seen in the course of the focus group that this view that the pandemic was an opportunity for the implementation of telerehabilitation in the country, not only for COVID-19 patients, but also for other chronic situations, was due to the realization of the advantages of telerehabilitation as a safe and effective method for patient and family support and continuity of care.


*I’ll also say that it’s really innovative, because in our area I don’t think it’s very common to do breathing exercises via video call. But what is certain is that we were able to perceive that we obtained good results in terms of diagnostic resolution of compromised breathing patterns, ineffective airway clearance, lack of knowledge, for example, of how inhalation therapy works. (P8)*



*Not having to bring patients in difficult conditions to the hospital, their issues could be solved at home. (P2)*



*Here we have this issue… the patients live far from the place and for them it is already a sacrifice… And so, we managed to grab these patients… hospital… and it has been a fantastic experience for us too. (P4)*


The participants listed and valued the advantages related to the people’s comfort, reduction in the number of trips, time savings with visits to health services, and the individualization of care. Their comments showed that a set of techniques, up to a certain level of complexity, can be carried out remotely, with professionals providing information, training, and supervision:


*It is also possible, through this method, to work on intolerance to activity and, as far as possible, implement strength training and resistance training by video consultation. (P8)*


The experience with telerehabilitation also allowed for the understanding that it is possible to offer the same care as in face-to-face rehabilitation programs and that, in this modality, interactions between professionals and patients remain possible, in good time and with the same quality. This could lessen the main area of distrust regarding this modality, which is that of not establishing relationships that allow individualized care.

## 4. Discussion

The COVID-19 pandemic was a challenge to health systems and their ability to respond. It is indisputable that to respond to the pandemic, the focus of most efforts was on hospital care compared to the attention given to the needs of all the patients who returned home [21]. We corroborate the opinion of some authors that, in the face of the pandemic, new questions were raised, both in controlling the spread of COVID-19 and in preparing for the safe transition of patients from hospitals to communities [11,13,14]. One of the successful responses to these challenges has been the development of a hierarchical system for patient identification and routing of clinical messages and virtual care models, which have been widely accepted by patients and represent a key component in providing safe and timely health care during the pandemic [22].

The participants noted that a major challenge for systems and professionals was coordination between levels of care. That is because of the complexity of the transitions experienced by patients after COVID-19, which are combined with issues related to multiple comorbidities, social and economic conditions, emotional stability, functional disability secondary to hospitalization, decreased ability in self-care, and changes in health status [21]. This results in a need for complex transitional care to control respiratory symptoms, prevent complications, and allow a return to pre-hospital admission levels of functionality.

In the perceptions of the specialists, initial ignorance about the disease and its sequelae, as well as the realization that the post-COVID-19 journey is long and implies multidimensional interventions at the emotional, cognitive, and functional levels [23], challenged healthcare facilities to develop new ‘ways of doing’ to guarantee continuity of care, such as through coordination between hospital nurses and nurses in primary health care.

Still in relation to the category of coordination between levels of care, there was consensus that feelings of fear and insecurity in the face of the unknown existed among professionals at all levels of care. These feelings were assuaged by learning about care while building knowledge about the disease, its prognosis, and rehabilitation. The challenges presented by difficulties experienced in planning the return home led to the emergence of interactions and coordination among, and integration of, all participants in care, ensuring discharge planning and safe subsequent follow-up at all stages of transitional care [24].

Understanding of the differences between the cultures of hospital care and community care revealed that care planning needs to be carried out according to individual needs, with a central focus on care, and not on the environments of care practices. In order to achieve this goal, communication and pre-discharge coordination are seen as a priority. They contribute to discussion about and negotiations of transitional care plans [24,25,26].

There was a consensus that coordination should be multidisciplinary, and that in the face of transitional care programs that guarantee continuity of interventions initiated in hospitals, nurse specialists in rehabilitation are the professionals who, within the scope of their competencies, are best prepared to manage transitional care [26], according to the competency framework for this area of specialization.

The results also highlighted that transitional care, in an e-health modality, allows for early communication and facilitates contact between home care teams and hospital teams. The participants noted that e-health is a tool that allows the early identification of post-COVID people’s needs; exchange of information about physical, behavioral, cognitive, and functional deficits and social vulnerability allows for early intervention in communities [11,13,24]. It promotes collaboration between teams in order to reach consensus on care plans among hospital and primary care team members. It facilitates meetings among all those involved in the transitional process, allowing them to work in teams that include staff from both primary health care and hospital health care. In this way, it is possible to guarantee the provision of timely and effective transitional care [24,26,27].

The adoption of e-heath is an inexorable part of the movement toward adaptation to global digitalization trends. Its evolution in recent years, especially during the COVID-19 pandemic, both in terms of technology and in terms of experiences of its implementation, has allowed for improving coordination between teams, shortening distances, and developing new projects that include telerehabilitation [22].

The results of this study support the assertion that telerehabilitation achieves results similar to those of face-to-face rehabilitation [22] with multiple clear advantages. It improves access to rehabilitation programs, with greater comfort and shortened distances, especially in inland areas of the country, where health services are distant from people’s homes [22,28]. It also enables complex respiratory rehabilitation programs, with the aim of improving functional capacity, dyspnea, and quality of life [13,29,30]. In addition, it can be carried out safely, with the possibility of telemonitoring or intervention in the presence of informal caregivers [29,30].

Also noteworthy is the observation of the clinical results of interventions, measured by technologies, in controlling dyspnea, improving functional capacity, qualifying people for self-care, and having favorable impacts on health-related quality of life [31]. As the experts in the focus groups said, e-health allows for personalized, continuous care. The identification of their own signs and symptoms, accompanied by health professionals, helps patients learn to deal with them, and also helps them become more informed and aware in decision-making, acquiring tools for the management of the physical, emotional, and social impacts of their health issues.

Telehealth, applied in a structured way to rehabilitation, allows the optimization of resources while maintaining the results that are achieved in person, with greater comfort and less risk, and promotes a more sustainable health system.

### Limitations

The limitations of this study stem from the method itself, since its success depends on trust in the researcher’s focus and group interactions. During the discussion, care was taken in posing questions related to the design of the investigation, allowing all interlocutors to express their opinions and ideas, seeking to reach depth in the subject. On the part of the moderator, care was taken not to influence the participants’ ideas or lead the discussion toward a specific conclusion. The intentional choice of participants and a single focus group limits the results to the context and does not allow inference for other contexts.

Despite these limitations, we believe that the findings allow us to answer the research question.

## 5. Conclusions

The COVID-19 pandemic accelerated the process of digitalization of health care, bringing with it a new wave of change, with an impact on the organization of health care. Telehealth allows proximity between hospital health teams and primary health care, and can be considered one instrument in the creation of transitional nursing care teams. It can act as a facilitator in communication and care coordination, with rehabilitation nurses being the most qualified professionals to be managers of transitional care.

As for telerehabilitation intervention as transitional care, face-to-face assessment by care-process participants is necessary in order to establish care plans. Face-to-face moments are important for the assessment of specific programs, considering respiratory rehabilitation programs, teaching, training, self-care training, and physical exercise training. Concern about the training of informal caregivers stood out, since in the context of the COVID-19 infection, in-person healthcare visits decreased.

Whenever necessary, e-health programs must resort to face-to-face visits, tailoring programs to people according to their needs. It became clear that e-heath in transitional care is an opportunity for rehabilitation nursing care. E-heath allows greater proximity to patients, decreases hospital visits, and reduces the risk of infections. Telerehabilitation allows exercise at home, and training in self-care in privacy for patients. It enables them to care for themselves. When digitally enabled and informed, they develop decision-making capacity.

There was a consensus among all the professionals that transitional care supported by telehealth in rehabilitation nursing care is a clear opportunity in the reorganization of care, the management of resources, and the control of infection.

The results of the present study can serve as an incentive for the education and training of health professionals in the use of the technology necessary for the implementation of telehealth programs, in particular, telerehabilitation.

It allowed reflection on the reorganization of transitional health care, opening space for the proposal of visits that use digital tools, ranging from primary health care to hospitals, in order to create therapeutic plans in transitional care

Health professionals should act as facilitators in transitional care through practices that place people and their caregivers at the center of care.

## Figures and Tables

**Figure 1 ijerph-19-17096-f001:**
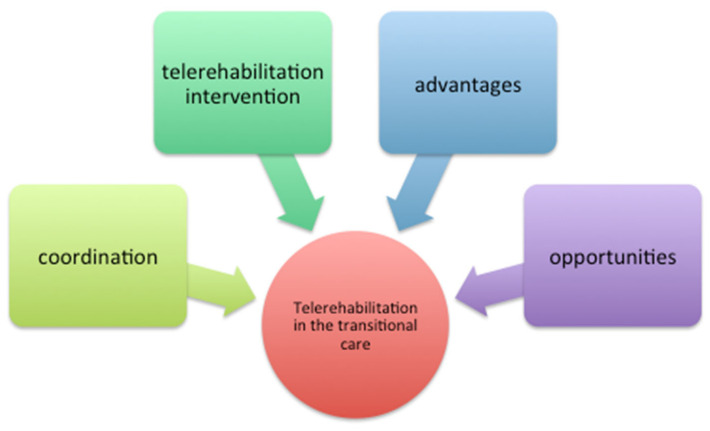
Relation between categories. Lisbon, 2022.

## Data Availability

The data used during this study are available from the corresponding author, under request by email.

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
