# Peer review of "Telerehabilitation in the Transitional Care of Patients with Sequelae Associated with COVID-19: Perception of Portuguese Nurses"

_ijerph, 2022, doi:10.3390/ijerph192417096_

Round 1
Reviewer 1 Report
The paper focuses on identifying aspects/components to consider in planning and implementing telerehabilitation interventions that ensure transitional care for people with long COVID-19 after hospitalization.
The structure of the work is clear and concise, establishing a focused objective and a methodology that is coherent to achieve it.
In that sense, the work, is of high relevance as it addresses a critical issue related to the transitional care of patients with long COVID-19 after long hospitalizations. Specifically related to the identification of factors and components to be considered in the planning and implementation of telerehabilitation interventions.
In the introduction, it would be essential to refer to transitional care, what it consists of, its definition, what is the purpose and function, if any, in its application in the context of COVID-19 patients. And its relationship with telerehabilitation.
In the methodology section, a diagram could be included where these variables can be considered and their interrelationships established. Health personnel, patients, transitional care, COVID-19 patients, etc. This same diagram could be used to organize the related outcomes and how to take them into consideration for telerehabilitation planning and implementation.
Other minor, but no less important details that should be corrected:
Revise keywords: Rransitional Care
Author Response
Dear reviewer:
Thank you for taking the time to review our article. The suggestions made helped to improve the quality of the article.
The paper focuses on identifying aspects/components to consider in planning and implementing telerehabilitation interventions that ensure transitional care for people with long COVID-19 after hospitalization.
The structure of the work is clear and concise, establishing a focused objective and a methodology that is coherent to achieve it.
In that sense, the work, is of high relevance as it addresses a critical issue related to the transitional care of patients with long COVID-19 after long hospitalizations. Specifically related to the identification of factors and components to be considered in the planning and implementation of telerehabilitation interventions.
In the introduction, it would be essential to refer to transitional care, what it consists of, its definition, what is the purpose and function, if any, in its application in the context of COVID-19 patients. And its relationship with telerehabilitation.
Done. We introduced 2 paragraphs in the end of introduction to define transitional care and it relationship with telerehabilitation.
In the methodology section, a diagram could be included where these variables can be considered and their interrelationships established. Health personnel, patients, transitional care, COVID-19 patients, etc. This same diagram could be used to organize the related outcomes and how to take them into consideration for telerehabilitation planning and implementation.
We added an figure in page 4 to illustrate the categories.
Other minor, but no less important details that should be corrected:
Revise keywords: Rransitional Care
We corrected it.
Reviewer 2 Report
This study aims to identify the aspects/components to be considered in the planning and implementation of telerehabilitation interventions that guarantee transitional care for people with long COVID-19 after hospitalization. This topic is valuable while the COVID-19 is highly infectious and pathogenic. There are several suggestions for this study.
1. A block diagram will better for readers understand the main contents of this study.
2. This paper mainly studies the expression of the 8 nurses in the focus group.
a) Are there more participants for analysis? Or for verification? To improve the persuasiveness of this article?
b) Is there any known virus or disease similar to COVID-19, which can provide a reference for this study?
In a word, the research materials in this paper are too few, resulting in a lack of persuasiveness.
3. The text of Section 3 (Results) is too long. Can it be further divided into several sections? This is more helpful for readers to find the key points. This concern also exists in Section 4 (Discussion).
Author Response
Answer to Review R2
Dear reviewer:
Thank you for taking the time to review our article. The suggestions made helped to improve the quality of the article.
- A block diagram will better for readers understand the main contents of this study.
We added an figure in page 4 to illustrate the categories.
- This paper mainly studies the expression of the 8 nurses in the focus group.
- a) Are there more participants for analysis? Or for verification? To improve the persuasiveness of this article?
Only a Focus Group with 8 participants was carried out to answer the research question. It should be noted that in Portugal, we do not have many professionals using telerehabilitation, which makes it difficult to have more participants. On the other hand, the emergence of the post-covid situation required a quick response to guide the practice of professionals.
- b) Is there any known virus or disease similar to COVID-19, which can provide a reference for this study?
No, because post covid injuries are different from the sequels of other respiratory diseases.
In a word, the research materials in this paper are too few, resulting in a lack of persuasiveness.
We have that notion.
- The text of Section 3 (Results) is too long. Can it be further divided into several sections? This is more helpful for readers to find the key points. This concern also exists in Section 4 (Discussion).
We added subtopics in the results to facilitate the read of this section. In discussion we didn’t the same because it is not usual in the journal to appear articles with the discussion divided/separated into subtopics.